# Gradient-based training of Gaussian Mixture Models in High-Dimensional Spaces

## Abstract

We present an approach for efficiently training Gaussian Mixture Models (GMMs) with Stochastic Gradient Descent (SGD) on large amounts of high-dimensional data (e.g., images). In such a scenario, SGD is strongly superior in terms of execution time and memory usage, although it is conceptually more complex than the traditional Expectation-Maximization (EM) algorithm. For enabling SGD training, we propose three novel ideas: First, we show that minimizing an upper bound to the GMM log likelihood instead of the full one is feasible and numerically much more stable way in high-dimensional spaces. Secondly, we propose a new annealing procedure that prevents SGD from converging to pathological local minima. We also propose an SGD-compatible simplification to the full GMM model based on local principal directions, which avoids excessive memory use in high-dimensional spaces due to quadratic growth of covariance matrices. Experiments on several standard image datasets show the validity of our approach, and we provide a publicly available TensorFlow implementation.

## 1 Introduction

This contribution is in the context of Gaussian Mixture Models (GMMs), which is a probabilistic unsupervised method for clustering and data modeling. GMMs have been used in a wide range of scenarios, e.g., Melnykov et al. (2010). Traditionally, free parameters of a GMM are estimated using the so-called *Expectation-Maximization* (EM) algorithm, which has the appealing property of requiring no learning rates and which automatically enforces all constraints that GMMs impose.

### 1.1 Motivation

GMMs have several properties that make their application to image data appealing, like the ability to generate samples from the modeled distribution, or their intrinsic outlier detection ability. However, for such high-dimensional and data-intensive scenarios which require a high degree of parallelization to be efficiently solvable, the traditional way of training GMMs by EM is reaching its limits, both in terms of memory consumption and execution speed. Memory requirements skyrocket mainly because GMM is intrinsically a batch-type algorithm that needs to store and process all samples in memory in order to be parallelizable. In addition, computation time becomes excessive because EM involves covariance matrix inversions, which become prohibitive for high data dimensionality. Here, an online or mini-batch type of optimization such as Stochastic Gradient Descent (SGD) has strong advantages w.r.t. the memory consumption in particular, but also w.r.t. computation time since it requires no matrix inversions. Therefore, we seek to develop a Stochastic Gradient Descent algorithm for GMMs, making it possible to efficiently train GMMs on large collections of images. Additionally, since the number of free GMM parameters depends quadratically on the data dimensionality $D$, we seek to develop simplified GMM models with a less severe impact of $D$.

### 1.2 Related Work

There are several proposals for what is termed "incremental" or "online" EM for training GMMs Vlassis & Likas (2002); Engel & Heinen (2010); Pinto & Engel (2015); Cederborg et al. (2010); Song & Wang (2005); Kristan et al. (2008), allowing GMMs to be trained by providing one sample at a time. However, none of these models reproduce the original GMM algorithm faithfully, which is

why most of them additionally resort to component pruning or merging heuristics for components, leading to learning dynamics that are difficult to understand. This is also the case for the Locally Weighted Projection Regression (LWPR) algorithm Vijayakumar et al. (2005) which can be interpreted as a GMM model adapted for online learning. To our knowledge, there is only a single work proposing to train GMMs by SGD Hosseini & Sra (2015). Here, constraint enforcement is ensured by using manifold optimization techniques and a carefully chosen set of regularizers, leading to a rather complex model with several additional and hard-to-choose hyper-parameters. The idea of including regularization/annealing into GMMs (in the context of EM) was proposed in Verbeek et al. (2005); Ormoneit & Tresp (1998), although their regularizes significantly differs from ours. Approximating the full GMM log-likelihood is used in Verbeek et al. (2003); Pinheiro & Bates (1995); Dognin et al. (2009), although together with EM training. There has been significant work on simplifying GMM models (e.g., Jakovljević (2014)), leading to what is called *parsimonious GMM models*, which is reviewed in Bouveyron & Brunet-Saumard (2014).

## 1.3 CONTRIBUTIONS

The main novel contributions of this article are:

- a generic and novel method for training GMMs using standard SGD that enforces all constraints and is numerically stable for high dimensional data (e.g., $D > 500$)
- an annealing procedure that ensures convergence from a wide range of initial conditions in a streaming setting (excluding, e.g., centroid initialization by k-means)
- a simplification of the basic GMM model that is particularly suited for applying GMMs to high-dimensional data
- an analysis of the link between SGD-trained GMM models and self-organizing maps

Apart from these novel contributions, we provide a publicly available TensorFlow implementation.[1]

## 2 DATASETS

We use two datasets for all experiments. **MNIST** (LeCun et al., 1998) consists of gray scale images of handwritten digits (0-9) and is a common benchmark for computer vision systems and classification problems. **SVHN** (Netzer et al., 2011) is a 10-class benchmark based on color images of house numbers (0-9). We use the cropped and centered digit format. For both benchmarks, we use either the full images ($32 \times 32 \times 3$ for SVHN, $28 \times 28 \times 1$ for MNIST), or a crop of the central $6 \times 6$ patch of each image. The class information is not used since GMM is a purely unsupervised method.

## 3 METHODS: STOCHASTIC GRADIENT DESCENT FOR GMMS

Gaussian Mixture Models (GMMs) are normally formulated in terms of $K$ component probabilities, modeled by multi-variate Gaussians. It is assumed that each data sample $\boldsymbol{x}$, represented as a single row of the data matrix $\boldsymbol{X}$, has been sampled from a single Gaussian component $k$, selected with a priori probability $\pi_k$. Sampling is performed from the Gaussian conditional probability $p(\vec{x}|k)$, whose parameters are the centroids $\boldsymbol{\mu}_k$ and covariance matrices $\boldsymbol{\Sigma}_k$: $p(\boldsymbol{x}|k) \sim \mathcal{N}(\boldsymbol{x}; \boldsymbol{\mu}, \boldsymbol{\Sigma})$. A GMM aims to explain the data by minimizing the log-likelihood function

$$\mathcal{L} = -\frac{1}{N} \sum_{n=1}^{N} \log \sum_{k} \pi_k p(\boldsymbol{x}_n|k). \tag{1}$$

Free parameters to be adapted are the component probabilities $\pi_k$, the component centroids $\boldsymbol{\mu}_k$ and the component covariance matrices $\boldsymbol{\Sigma}_k$.

### 3.1 MAX-COMPONENT APPROXIMATION

As a first step to implement Stochastic Gradient Descent (SGD) for optimizing the log-likelihood (see Eq. 1), we observe that the component weights $\pi_k$ and the conditional probabilities $p(\boldsymbol{x}|k)$ are

---

[1] https://github.com/anonymous-iclr20/GMM.git

positive by definition. It is therefore evident that any single component of the inner sum over the components $k$ is a lower bound for the whole inner sum. The largest of these $K$ lower bounds is given by the maximum over the components, thus giving

$$\mathcal{L} = -\frac{1}{N}\sum_{n=1}^{N}\log\sum_{k}\pi_k p(\boldsymbol{x}_n|k) \leq \mathcal{L}_{\text{MC}} = -\frac{1}{N}\sum_{n}\log\max_{k}\big(\pi_k p(\boldsymbol{x}_n|k)\big)$$
$$= -\frac{1}{N}\sum_{n}\max_{k}\log\big(\pi_k p(\boldsymbol{x}_n|k)\big). \tag{2}$$

This is what we term the *max-component approximation* of Eq. 2. Since $\mathcal{L}_{\text{MC}} \geq \mathcal{L}$, we can decrease $\mathcal{L}$ by minimizing $\mathcal{L}_{\text{MC}}$. The advantage of $\mathcal{L}_{\text{MC}}$ is that it is not affected by numerical instabilities as $\mathcal{L}$ is. An easy way to see this is by considering the case of a single Gaussian component with weight $\pi_k = 1$, centroid $\boldsymbol{\mu}_k = 0$ and whose covariance matrix $\boldsymbol{\Sigma}_k$ has only diagonal entries which are all equal to 0.1. In this case, an $1\,000$-dimensional input vector $\boldsymbol{x}$, $\boldsymbol{x}_i = 1\ \forall i$ would give a conditional probability of $p(\boldsymbol{x}|k) = (2\pi 0.1)^{-500}e^{-5000}$ which will, depending on floating point inaccuracy, produce NaN or infinity for the first factor due to overflow, and zero for the second factor due to underflow, resulting in an undefined behavior.

## 3.2 Constraint Enforcement

GMMs require the weights to be normalized: $\sum_k \pi_k = 1$ and elements of the diagonal matrices $\Sigma$ to be non-negative: $\Sigma_{ij} \geq 0\ \forall i, j$. In traditional GMM training via Expectation-Maximization, these constraints are taken care of by the method of Lagrangian multipliers. In SGD, they must be enforced in different manner. For the weights $\pi_k$, we adopt the approach proposed in Hosseini & Sra (2015), which replaces them by other free parameters $\xi_k$ from which the $\pi_k$ are computed such that normalization is ensured. One such computational scheme is

$$\pi_k = \frac{\exp(\xi_k)}{\sum_j \exp(\xi_j)}. \tag{3}$$

For ensuring non-negativeness of the covariances, we identify all covariances inferior to a certain minimal value $\Sigma^{\min}$ after each gradient descent step, and subsequently clip them to $\Sigma^{\min}$.

## 3.3 Annealing procedure

A major issue for SGD optimization of GMMs are local minima that obviously correspond to undesirable solutions. Prominently among these is what we term the *single-component solution*: here, a single component $k^*$ has a weight close to 1, with its centroid and covariance matrix being given by the mean and covariance of the data:

$$\pi_{k^*} \approx 1, \quad \boldsymbol{\mu}_{k^*} = \mathbb{E}[\boldsymbol{X}], \quad \boldsymbol{\Sigma}_{k^*} = \text{Cov}(\boldsymbol{X}). \tag{4}$$

Another undesirable local minimum is the *degenerate solution* where all components have the same weight, centroid and covariance matrix, the latter being identical to the single-component solution. See Fig. 4 (left) for a visualization of the latter.

Such solutions usually evolve from an incorrect initialization of the GMM model, and it is easy to show that the gradient w.r.t. all model parameters is zero in both of these scenarios. This is easy to show: As the gradients read (with $\gamma_{nl}$ denoting the responsibility of component $l$ for sample $n$)

$$\frac{\partial\mathcal{L}}{\partial\mu_{lm}} = \frac{1}{N}\sum_{n}\gamma_{nl}\pi_l\left(x_{nm} - \mu_{lm}\right) \tag{5}$$

$$\frac{\partial\mathcal{L}}{\partial\Sigma_{lm}} \sim \frac{1}{N}\sum_{n}\gamma_{nl}\pi_l\left(-\frac{2}{\Sigma_{nl}} + \frac{(x_{nm} - \mu_{lm})^2}{\Sigma_{lm}^3}\right), \tag{6}$$

it is easy to see that when either all or single centroid/covariance matrix are equal to the total mean and variance of the data ($\mu_{lm} = \frac{1}{N}x_{nm}$ and $\Sigma_{lm} = \frac{2}{N}(x_{nm} - \mu_{lm})^2$), we obtain zero gradients for degenerate solutions ($\pi_i = \gamma_i = 1/K$) and single-component solutions ($\exists! \pi_{i^*}, \gamma_{i^*} = 1$), respectively.

Our approach for avoiding these undesirable solutions in principle is to punish their characteristic response patterns by an appropriate modification of the loss function that is minimized, i.e., $\mathcal{L}_{\text{MC}}$. The following guidelines should be adhered to:

- single-component and degenerate solution should be punished by the annealing procedure
- as we mainly wish to ensure a good starting point for optimization, annealing should be active only at the beginning of the training process
- as training advances, annealed loss should smoothly transition into the original loss
- the number of free parameters introduced by the annealing procedure should be low

These requirements are met by what we term the *smoothed max-component log-likelihood* $\tilde{\mathcal{L}}_{\text{MC}}$ :

$$\tilde{\mathcal{L}}_{\text{MC}} = \frac{1}{N} \sum_n \max_k \left( \sum_j \boldsymbol{g}_{kj} \log \left( \pi_j p(\boldsymbol{x}_n|j) \right) \right). \tag{7}$$

Here, we assign a normalized coefficient vector $\boldsymbol{g}_k$ to each Gaussian mixture component $k$. The entries of $\boldsymbol{g}_k$ are computed in the following fashion:

- Assume that the $K$ Gaussian components are arranged on a 1D grid of dimensions $(1, K)$ or on a 2D grid of dimensions $(\sqrt{K}, \sqrt{K})$. Each linear component index $k$ has thus an unique associated 1D or 2D coordinate $\boldsymbol{c}(k)$.
- Assume that the vector $\boldsymbol{g}_k$ of length $K$ is actually representing a 1D structure of dimension $(1, K)$ or a 2D structure of dimension $(\sqrt{K}, \sqrt{K})$. Each linear vector index $j$ in $\boldsymbol{g}_k$ has thus a unique associated 1D or 2D coordinate $\boldsymbol{c}(j)$.
- The entries of the vector $\boldsymbol{g}_k$ are computed as

$$g_{kj} = \exp \left( -\frac{(\boldsymbol{c}(j) - \boldsymbol{c}(k))^2}{2\sigma^2} \right) \tag{8}$$

and subsequently normalize to have unit sum. Thus, Eq. 7 essentially represents a convolution of the probabilities $p(\vec{x}|k)$, arranged in an 1D or 2D grid, with a Gaussian convolution filter, thus amounting to a smoothing operation. The 1D or 2D variance $\sigma$ in Eq. 8 is a parameter that must be set as a function of the grid size such that Gaussians are neither homogeneous nor delta peaks. Thus, the loss function in Eq. 7 is maximized if the log probabilities follow an unimodal Gaussian profile of variance $\sigma$, whereas single-component and degenerate solutions are punished.

It is trivial to so see that the annealed loss function in Eq. 7 reduces to the non-annealed form Eq. 2 in the limit where $\sigma \to 0$. This is because the vectors $\boldsymbol{g}_k$ approach Kronecker deltas in this case, with only a single entry of value 1, which essentially removes the inner sum in Eq. 7. By making $\sigma$ time-dependent, starting at an intermediate value of $\sigma(t_0) \equiv \sigma_0$ and then let it approach a small final value $\sigma(t_\infty) \equiv \sigma_\infty$, we can smoothly transition the annealed loss function Eq. 7 to the original max-component log-likelihood Eq. 2. Time dependency of $\sigma(t)$ can thus be chosen to be:

$$\sigma(t) = \begin{cases} \sigma_0 & t < t_0 \\ \sigma_\infty & t > t_\infty \\ \sigma_0 \exp(-\tau t) & t_0 < t < t_\infty \end{cases} \tag{9}$$

where the time constant in the exponential is chosen as $\tau = \log \frac{\sigma_0 - \sigma_\infty}{t_\infty - t_0}$ to ensure a smooth transition.

## 3.4 FULL GMM MODEL TRAINED BY SGD AND ITS HYPER-PARAMETERS

Putting everything together, training Gaussian Mixture Models with SGD is performed by minimizing the smoothed max-component log-likelihood of Eq. 7, enforcing the constraints on the component weights and covariances as detailed in Sec. 3.2 and transitioning from the smoothed to the "bare" loss function as detailed in Sec. 3.3. In order to avoid convergence to other undesirable local minima by SGD, we introduce three weighting constants $\lambda_{\boldsymbol{\pi}}$, $\lambda_{\boldsymbol{\mu}}$ and $\lambda_{\boldsymbol{\Sigma}}$, all in the $[0, 1]$ range, for controlling the relative adaptation speeds for the three groups of free parameters in the GMM models

(setting, e.g., $\lambda_{\boldsymbol{\pi}} = 0$ would disable the adaptation of weights). We have thus the update rules:

$$\xi_k(t+1) = \xi_k(t) - \epsilon \cdot \lambda_{\boldsymbol{\pi}} \frac{\partial \tilde{\mathcal{L}}_{MC}}{\partial \xi_k},$$

$$\boldsymbol{\mu}_k(t+1) = \boldsymbol{\mu}_k(t) - \epsilon \cdot \lambda_\mu \frac{\partial \tilde{\mathcal{L}}_{MC}}{\partial \boldsymbol{\mu}_k} \text{ and} \quad (10)$$

$$\boldsymbol{\Sigma}_k(t+1) = \boldsymbol{\Sigma}_k(t) - \epsilon \cdot \lambda_\Sigma \frac{\partial \tilde{\mathcal{L}}_{MC}}{\partial \boldsymbol{\Sigma}_k}.$$

Centroids are initialized to random values in range $[-\mu^i, \mu^i]$, weights are chosen equiprobable, that is to say $1/K$, for all components, and covariance matrix entries are uniformly initialized to very small positive values $\Sigma^i$ (a good ad-hoc choice is $0.15$). Last but not least, a learning rate $\epsilon$ is required for performing SGD. We find that our algorithm depends very weakly on $\epsilon$, and that a fixed value of $\epsilon = 0.01$ is usually a good choice. We summarize good practices for choosing hyper-parameters of the proposed SGD approach for GMMs in App. C. Please not that it is not required to initialize the component centroids by k-means as it is usually recommended when training GMMs by EM.

### 3.5 SIMPLIFICATIONS AND MEMORY/PERFORMANCE ISSUES

For the full GMM model, regardless of whether it is trained by EM or SGD, the memory requirements of a GMM with $K$ components depend strongly on the data dimensionality $D$. A good measure for memory requirements is the number of free model parameters $M$ computed as $M = K(D^2 + D + 1)$. To reduce this number, we propose several strategies that are compatible with training by SGD (see also Fig. 1):

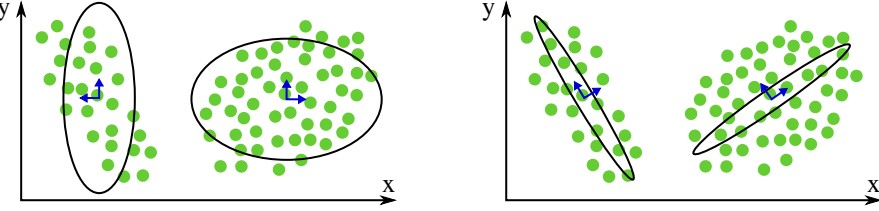

Figure 1: Two possibilities for simplifying GMMs, illustrated for a dataset with two clusters and two Gaussian components. Green dots represent data samples and arrows represent normalized local principal directions along which variances (represented by ellipses) are computed. Left: diagonal covariance matrix, where it is assumed that clusters vary only in the coordinate directions (not the case here). Right: non-diagonal covariance matrix where only a limited number (here one) of principal directions is considered for variance computation.

**Diagonal Covariance Matrix** A well-known simplification of GMMs consists of using a diagonal covariance matrix instead of a full one. This reduces the number of trainable parameters to $K(2D+1)$, where $D$ is the dimensionality of the data vectors and $K$ the number of components.

**Local Principal Directions** A potentially very powerful simplification consist of using a diagonal covariance matrix of $S < D$ diagonal entries, and letting SGD adapt the *local principal directions* along which these variances are measured for each component (in addition to the other free parameters, of course). The expression of the conditional component probabilities now includes, for each component $k$, the $S$ normalized principal direction vectors $\boldsymbol{d}_{ks}$, $s = 1, \dots, S$ and reads:

$$p(\boldsymbol{x}|k) = \frac{1}{\sqrt{(2\pi)^S \prod_{s=1}^S \Sigma_{ss}}} \exp\left(-\frac{1}{2} \sum_{s=1}^S \frac{\left(\boldsymbol{x}^T \boldsymbol{d}_{ks}\right)^2}{\Sigma_{ss}^2}\right). \quad (11)$$

This leads to a value of $M = K((S+1)D + 1)$, and thus achieves satisfactory model simplification only if $S$ can be chosen very small and $D$ is high. In order for the model to be fully determined, another constraint needs to be enforced after each SGD step, namely the orthogonality of the local principal directions for each component $k$: $\boldsymbol{p}_{kl}^T \boldsymbol{p}_{km} = \delta_{lm} \ \forall \, l, m < S$. For now, the matrix $\boldsymbol{D}_k$ whose rows are the $S$ vectors $\boldsymbol{d}_{ks}$ is subjected to a QR decomposition $\boldsymbol{D}_k = \boldsymbol{Q}_k \boldsymbol{R}_k$ and setting $\boldsymbol{D}_k = \boldsymbol{Q}_k$.

## 4 EXPERIMENTS

Unless otherwise stated, the experiments in this section will always be conducted with the following parameter values (see App. C for a justification of these choices): total iterations $T = 1\,000 \cdot \sqrt{K}$, $t_0 = 0.3T$, $t_\infty = 0.8T$, mini-batch size $B = 1$, $\mu^i = 0.01$, $\sigma_0 = 1.2$, $\sigma_\infty = 0.011$, $\epsilon = 0.011$, $\Sigma^{\min} = 0.15$ and $K = 5 \cdot 5$. The adaptation strengths $\lambda_\pi$, $\lambda_\mu$ and $\lambda_\sigma$ are all set to $1.0$. Except for Sec. 3.5, we will use a diagonal covariance matrix for each of the $K$ components. Training/test data are taken from MNIST or SVHN, either using full images or the central $6 \times 6$ patches.



Figure 2: Exemplary results for learned centroids, using normal experimental conditions, trained on variants of the two datasets (from left to right): MNIST, MNIST patches, SVHN, SVHN patches.

### 4.1 VALIDITY OF THE MAX-COMPONENT APPROXIMATION AND COMPARISON TO EM

Since we minimize only an approximation to the GMM log-likelihood, a first important question is about the quality of this approximation. To this end, we plot both energy functions over time when training on full MNIST and SVHN images, see Fig. 3. This figure shows that the log-likelihood obtained by EM-based training (using the implementation of *sklearn*) with the same configuration (using 10 iterations and a k-means-based initialization) achieves a very similar log-likelihood, which confirms that SGD is indeed comparable in performance.

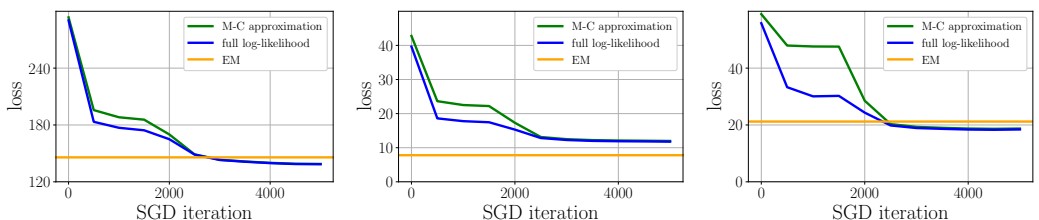

Figure 3: Comparing the max-component approximation to the negative log-likelihood during SGD training, shown for MNIST (left), MNIST patches (center) and SVHN patches (right).

### 4.2 EFFECTS AND BENEFITS OF ANNEALING

In order to demonstrate the beneficial effects of the annealing procedure, we conduct the basic experiment on full MNIST with annealing effectively turned off, which is achieved by setting $\sigma_0 = \sigma_\infty + 0.001 = 0.011$. As a result, in Fig. 4 we observe typical single-component solutions when visualizing the centroids and weights, along with a markedly inferior (higher) loss value of $\approx 155$, in contrast to $\approx 138$ when annealing is turned on. As a side effect, annealing enforces a topological ordering of centroids in the manner of a SOM (see also App. B for a mathematical analysis), which is only of aesthetic value for the moment, but might be exploited in the future.

### 4.3 BASIC FEASIBILITY AND PARAMETER SPACE

Here, we put everything together and train an GMM using SGD on various datasets of very high and intermediate dimensionality (results: see Fig. 2 and App. D). In particular, we analyze how the choice of the relative adaptation strengths $\lambda_\pi$, $\lambda_\Sigma$ and number of components $K$ affects the final log-likelihood. The results presented in Tab. 1 suggest that increasing $K$ is always beneficial, and

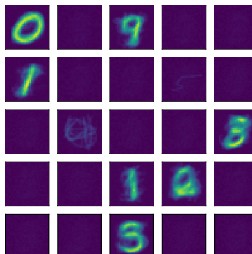 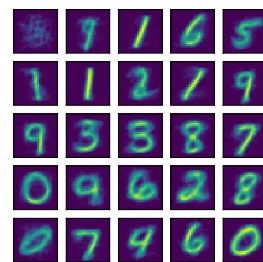

Figure 4: Effects of annealing on SGD convergence to centroids on MNIST. Left: single-component solution (no annealing). Right: regular solution (with annealing).

that the covariances and weights should be adapted more slowly than the centroids. The batch size $B$ had an effect on convergence speed but no significant impact on the final loss value (not shown here).

Table 1: Results for various hyper-parameter values, always given as the final max-component log-likelihood $\mathcal{L}_{\mathrm{MC}}$. The full log-likelihood values were always within $0.5$ of the given value, and smaller by construction. Each cell contains a value for the MNIST and one for the SVHN dataset.

| Parameter | $K=25$ | $K=36$ | $K=49$ | $K=64$ |
|---|---|---|---|---|
| $\lambda_\pi, \lambda_\Sigma = 1.0,\quad B=1$ | 138.1, 735.2 | 136.3, 704.7 | **133.3**, 693.5 | 134.1, **682.9** |
| $\lambda_\pi, \lambda_\Sigma = 0.1,\quad B=1$ | 138.3, 715.6 | 134.6, 706.5 | 133.5, **691.3** | **130.1**, 691.4 |
| $\lambda_\pi, \lambda_\Sigma = 0.0,\quad B=1$ | 138.1, 722.5 | 135.5, 714.3 | 133.2, 689.1 | **130.8**, **688**.4 |

## 4.4 ROBUSTNESS TO INITIAL CONDITIONS

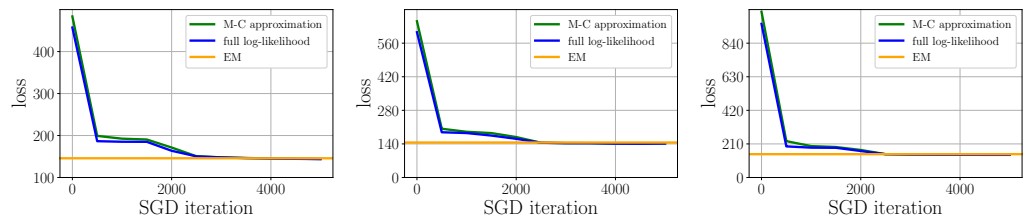

Figure 5: Comparing different initialization ranges for centroids: $[-0.5, 0.5]$ (left), $[-0.7, 0.7]$ (center) and $[-1, 1]$ (right). The orange line represents the same value in all diagrams, the seeming difference is due to scaling as initial loss values in some experiments are much higher (i.e., worse).

To test whether SGD converges as a function of the initial conditions, we train three GMMs on MNIST using three different initializations: $\mu^i \in \{1.0, 0.7, 0.5\}$. All of these choices led to non-convergence of the EM-based GMM implementation of *sklearn*. Differently from the other experiments, we choose a learning rate of $\epsilon = 0.02$ since the original value does not lead to convergent learning. The alternative is to double the training time which works as well. Fig. 5 shows the development of the loss function in each of the three cases, and we observe that all three cases indeed converge by comparing the final loss values to Fig. 3 (middle) and to the EM baseline. This behavior persists across all datasets, but only MNIST is shown due to space limitations.

## 4.5 SGD TRAINING OF SIMPLIFIED GMMS WITH PRINCIPAL LOCAL DIRECTIONS

We train a GMM with $S \in \{26, 50, 100\}$ local principal directions on MNIST and SVHN patches and log the final loss function values in order to assess model quality. We find that learned centroids and final loss values approach those of the diagonal model for $S \geq 20$ for both datasets. Please see App. A for a visualization of the final centroids as a function of $S$. For the full datasets, we find that

MNIST and SVHN require values of $S \geq 100$ and $S \geq 300$, respectively for achieving similar loss values as the diagonal model.

## 5 DISCUSSION

We showed that GMMs can be trained on high-dimensional image data by SGD, at very small batch sizes, in a purely online fashion. Memory and execution time requirements are very modest, and can be reduced even more by using intelligent simplifications of the GMM model. Here, we would like to discuss a few noteworthy points and suggest some avenues for improving the model:

**Robust Convergence** Unlike EM approaches which require a careful selection of starting point, the presented SGD scheme is very robust to initial conditions due to the proposed annealing scheme. In all experiments with high-dimensional image data, we found no initializations that did not, given reasonable time, converge to a regular solution.

**Numerical Stability** While we are not optimizing the full log-likelihood here, the max-component approximation is actually a good one, and has the advantage of absolute numerical stability. Clearly, clever ways might be found to avoid instabilities in the log-likelihood function computation, but when it comes to automatically computed gradients this is no longer possible, and indeed we found that it was mainly the (inaccessible) gradient computations where NaN values happened mostly.

**Complex annealing procedure** The complex way of regularizing the model, which is similar in spirit to simulated annealing approaches, may seem daunting, although the time constants can be determined by simple heuristics, see App. C. A next step will be to replace this procedure by an automated scheme based on an evaluation of the loss function: when it is stationary, $\sigma$ can be decreased slightly. Repeating this strategy until a small minimal value is reached should remove the need to fix annealing parameters except for time constants which can be chosen as a function of dataset size.

**Free Parameters** Apart from the annealing mechanism, the model contains the same parameters any SGD approach would, the learning rate $\epsilon$ and the weighting constants for the model parameters: $\lambda_\pi$, $\lambda_\mu$ and $\lambda_\Sigma$. We found no experimental evidence suggesting that the latter require values different from 1, although from general principles it might be sensible to adapt the variances and weights slower than the centroids. Further, it is required to understand how these constants can be used to obtain better solutions. Interesting is that standard DNN optimizers like Adam Kingma & Ba (2014) did not seem to produce satisfactory solutions which also requires looking into.

**Approximate Gradient Descent** The procedure described in Sec. 3.4 performs gradient descent on an approximation to the full log-likelihood Eq. 1. This might seem unsatisfactory, so we would like to point out that the original EM procedure does nothing else: it optimizes a lower bound of the log-likelihood. In particular, the M-step is not guaranteed to improve the log-likelihood at all: all we know is that it will never make it worse. In our SGD approach, there is a simple way to fix this shortcoming, which we will tackle as an next step: we can replace the max-operation in Eq. 2 by a softmax function with an initially large steepness parameter $\omega$ which will make it indistinguishable from a discrete max operation. After convergence, $\omega$ can be relaxed, which will result in a smooth transition to optimizing the full log-likelihood, but from an already well-converged state.

## 6 CONCLUSION AND OUTLOOK

On a more abstract level, we developed the presented method because of our interest in continual or incremental learning, which is essentially about eliminating the well-known catastrophic forgetting effect. We believe GMMs are an essential building block for continual learning models since their parameter updates are purely local in the sense that only components that are close to the currently best-matching component are updated. Next steps will consist of constructing deep network classifiers from convolutional GMM layers, with a readout layer on top (all layers being trained by SGD). Further, we will investigate how to sample efficiently from such hierarchical convolutional GMMs, allow generating large batches of samples in a replay-based architecture for continual learning.

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

## A    LOCAL PRINCIPAL DIRECTIONS

Here, we give a visualization of the resulting centroids for various choices of $S$ when training on MNIST patches.

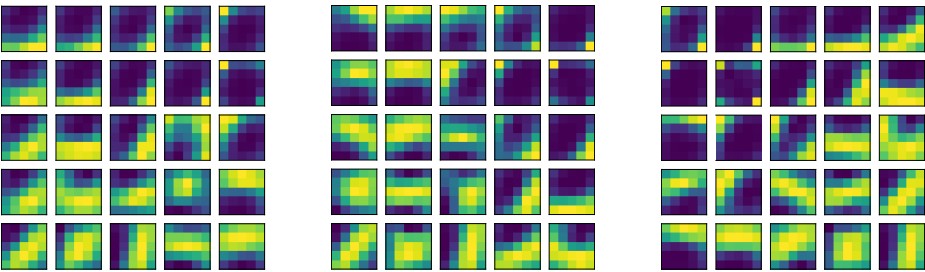

Figure 6: Comparing different values of $S$ when training GMMs with the local principal directions simplification. Shown are centroids for $S = 25$ (left), $S = 50$ (middle) and $S = 100$ (right).

## B    RIGOROUS LINK TO SELF-ORGANIZING MAPS

It is known that the self-organizing map (SOM, Kohonen (1990)) rule has no energy function it is minimizing. However, some modifications (see Heskes (1999)) have been proposed that ensure the existence of a $C^\infty$ energy function. These energy-based SOM models reproduce all features of the original model and use a learning rule that the original SOM algorithm is actually approximating very closely. In the notation of this article, SOMs model the data through $K$ prototypes $\boldsymbol{\mu}_k$ and $K$ neighborhood functions $\boldsymbol{g}_k$ defined on a periodic 2D grid, and their energy function is written as

$$\mathcal{L}_{SOM} = \frac{1}{N} \sum_n \max_k \sum_j g_{kj} \|\boldsymbol{x}_n - \boldsymbol{\mu}_k\|^2. \tag{12}$$

Discarding constant terms, we find that this is actually identical to the max-component log-likelihood approximation given in Eq. 2 with constant equiprobable weights $\pi_k$ and a constant diagonal $\boldsymbol{\Sigma}$ with equal diagonal entries. To our knowledge, this is the first time that a rigorous link between SOMs and GMMs has been established based on a comparison of energy functions, showing that SOMs are actually implementing a annealing-type approximation to the full GMM model, with component weights and variances chosen in a special way.

## C    RULES-OF-THUMB FOR SGD-TRAINING OF GMMS ON IMAGES

When training DNNs by SGD, several parameters need to be set according to heuristics. Here, we present some rules-of-thumb for performing this selection with GMMs. Generally, we can always set the batch size to 1, $\Sigma^{\min} = 0.15$ and $\epsilon = 0.01$. In all of our experiments, it was always feasible to set $t_0$ to half an epoch, and $t_\infty - t_0$ to half an epoch, too. However, tuning these parameters can speed up training significantly. The number of prototypes is a critical choice, but follows a simple "the more the better" rule. From the mathematical foundations of GMMs, it is evident that more components must always be able to reach a lower loss (except for possible local minima). The relative adaptation strengths $\lambda_\pi$, $\lambda_\Sigma$ and $\lambda_\mu$ can always be set to 1 (at least we did not find any examples where that did not work).

1. Choose the number of iterations a an increasing function of $K$: the more components, the more free parameters, the more time to train is needed

2. Flatten images, and put them as rows with random indices (as usual for SGD) into a tensor $\boldsymbol{X}$

3. Choose a quadratic number of components as $n^2$

4. Initialize $\sigma_0$ to $n/3$.

5. Preliminary train the GMM with constant $\sigma_0$. Select $t_0$ as the first time step where loss does no longer decrease. It never hurts if $t_0$ is chosen too large, it just takes longer.

6. Select $t_\infty$ by trial and error on the train set: start with very large values and use the smallest value that gives the same final loss value. Larger values for $t_\infty$ never hurt but take longer.

7. Train with the chosen parameters!

# D    MORE RESULTS

Visualization of prototypes training on different datasets (extracted 10 random classes, see Pfülb & Gepperth (2019) for details about the datasets). Each result (in the form of centroid visualizations) is recorded after training for $\frac{1}{4}$ epoch on a given dataset with $K = 25$ (see Figure 7 and Figure 8).

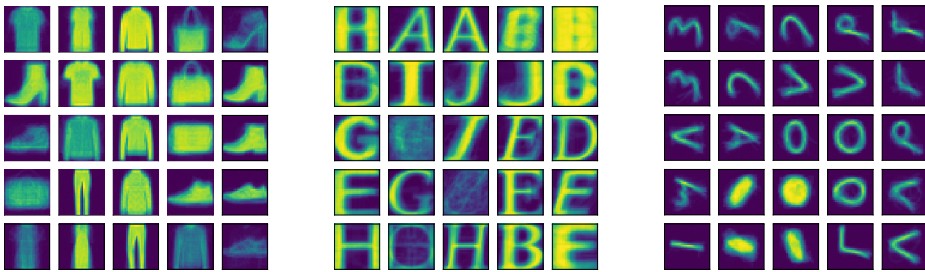

Figure 7: Visualization of prototypes training on different datasets (from left to right): FashionM-NIST (15000 iterations), NotMNIST (132275 iterations) and MADBase (15000 iterations).

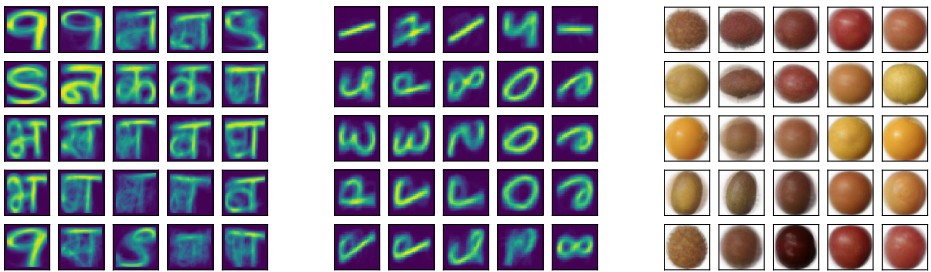

Figure 8: Visualization of prototypes training on different datasets (from left to right): Devanagari (4500 iterations), EMNIST (86250 iterations, rotated and mirrored MNIST classes) and Fruits (1475 iterations).

