# OpenReview forum: "Gradient-based training of Gaussian Mixture Models in High-Dimensional Spaces"
_ICLR.cc/2020/Conference — Reject_

### Official Review · AnonReviewer1 · 2019-10-23
**Official Blind Review #1**

**Rating:** 3

**Review:**

The paper describes in detail a proper implementation of SGD for learning GMMs. GMMs are admittedly one of the basic models in unsupervised learning, so the topic is relevant to ICLR, even though they are not particularly a hot topic.

The paper is overall clear and well-written. The main contributions are an effective learning of GMMs from random initialization that is competitive (in terms of final loss) to EM training with K-means initialization. The authors discuss a max-component loss instead of standard likelihood for numerical stability, using a softmax reparametrization for component weights to ensure they are in the simplex, and an annealed smoothing of the max-likelihood based on arbitrarily embedding components indexes on a 2D regular grid. Experiments are shown on MNIST and SVHN comparing different hyper parameter settings to a baseline EM implementation from scikit-learn.

- max-component: the use of the log(max p_k) instead of log(sum_k p_k) is sufficiently motivated to avoid well-known numerical problems that arise from directly computing the p_k rather than in log space. However, it is also a standard trick (e.g., in logsoftmax computations) to compute log(sum_k p_k) from log-probabilities using (for k* in argmax_k p_k):

log(sum_k p_k) = log(p_k*) + log(sum_k exp(log p_k - log p_k*))

which is numerically more stable. Maybe I'm missing something, but I do not understand the advantage of the max-component compared to this, so it seems to me that the max-component trick is a fairly limited contribution in itself.

- the main novelty seems to be the regularizer. The authors present experiments to show the effectiveness of it to avoid single-component solutions that may arise from random initialization, which is an interesting point of the paper. The motivation for smoothing on the 2D grid is still somewhat mysterious to me, even though the relationship with SOMs of Appendix B is interesting.

- the paper falls a bit short in showing any improvement compared to other baselines. The experiments describe some ablations, but does not really answer some questions: is regularization important if one uses K-means initialization? are there any practical advantages (e.g. learning speed, hyperparamter tuning) compared to standard EM? The authors say that the selection of the starting point in EM is important. This is a fair point, but it also seems solved by K-means. While the authors describe a "tutorial" for choosing the hyper parameters of their method, it still seems fairly manual. So the practical advantage of the method (which probably exists) would benefit from more comparisons.

- one of the difficulties in training GMMs comes from learning covariance matrices. While the authors discuss some way to train low-rank models, the only successful results seem to be with diagonal covariance matrices, which seems much easier. For instance, at least a toy example in which the low-rank version is useful would be interesting.

Overall, it seems to me that the work is serious, and describes possibly interesting how-tos for training GMMs. The main contribution is to describe a simple method to learn GMMs from random initialization. The main technical novelty seems the regularizer, which seems to work. The method has the advantage of simplicity, but successful results have only been shown with diagonal covariance matrices and it is unclear exactly what is gained over EM+K-means initialization.


other comments:
- negative log-likelihood is used in the results section. It would be good to clarify it somewhere since the paper only mentions "log-likelihood" but report a loss that should be minimized

- Section 4.4 "Differently from the other experiments, we choose a learning rate of  = 0.02 since the original value does not lead to convergent learning. The alternative is to double the training time which works as well." -> in the first sentence, I suppose it is more a matter of "slow learning" than "non-convergent learning"



**Experience Assessment:**

I have read many papers in this area.

**Review Assessment: Checking Correctness Of Derivations And Theory:**

I assessed the sensibility of the derivations and theory.

**Review Assessment: Checking Correctness Of Experiments:**

I assessed the sensibility of the experiments.

**Review Assessment: Thoroughness In Paper Reading:**

I read the paper thoroughly.

---

> ### Author Response · Authors · 2019-11-15
> **Response to reviewers**
>
> Thank you for the comprehensive review, it is the first time we publish on GMMs so this feedback is invaluable. Here our responses, we could not incorporate all the changes yet but they will come!
>
> - max-component: the use of the log(max p_k) instead of log(sum_k p_k) is sufficiently motivated to avoid well-known numerical problems that arise from directly computing the p_k rather than in log space. However, it is also a standard trick (e.g., in logsoftmax computations) to compute log(sum_k p_k) from log-probabilities using (for k* in argmax_k p_k): log(sum_k p_k) = log(p_k*) + log(sum_k exp(log p_k - log p_k*)) which is numerically more stable. Maybe I'm missing something, but I do not understand the advantage of the max-component compared to this, so it seems to me that the max-component trick is a fairly limited contribution in itself.
> **RESPONSE**  We are aware of this trick. It is ok for EM where it cures numerical instabilities by normalizing by the max. If we do this for the full gradient of the log-likelihood (which is different from the simplified EM form), the numerical instabilities are not cured.
>
> - the main novelty seems to be the regularizer. The authors present experiments to show the effectiveness of it to avoid single-component solutions that may arise from random initialization, which is an interesting point of the paper. The motivation for smoothing on the 2D grid is still somewhat mysterious to me, even though the relationship with SOMs of Appendix B is interesting.
> **RESPONSE** the motivation comes actually from the SOM domain. Maybe the term "regularizer" is misleading, it is actually more akin to annealing, with the same goal of avoiding spurious local minima.
>
> - the paper falls a bit short in showing any improvement compared to other baselines. The experiments describe some ablations, but does not really answer some questions: is regularization important if one uses K-means initialization? are there any practical advantages (e.g. learning speed, hyper-parameter tuning) compared to standard EM? The authors say that the selection of the starting point in EM is important. This is a fair point, but it also seems solved by K-means.
> **RESPONSE** we wish to use these SGD-based GMMs in streaming settings, where the entirety of data is not known in advance.  We will clarify this.
>
> While the authors describe a "tutorial" for choosing the hyper parameters of their method, it still seems fairly manual. So the practical advantage of the method (which probably exists) would benefit from more comparisons.
>
> - one of the difficulties in training GMMs comes from learning covariance matrices. While the authors discuss some way to train low-rank models, the only successful results seem to be with diagonal covariance matrices, which seems much easier. For instance, at least a toy example in which the low-rank version is useful would be interesting.
> **RESPONSE** we will try to add one. The point of the principal directions approach is that one effectively no longer uses a diagonal covariance matrix, but an approximation to the full one since the local directions of maximal covariance are learned from data.
>
>  Overall, it seems to me that the work is serious, and describes possibly interesting how-tos for training GMMs. The main contribution is to describe a simple method to learn GMMs from random initialization. The main technical novelty seems the regularizer, which seems to work. The method has the advantage of simplicity, but successful results have only been shown with diagonal covariance matrices and it is unclear exactly what is gained over EM+K-means initialization.
> **RESPONSE** the main point (that wasn't maybe stated as clearly as it could have been) is that we wish to perform SGD in order to use GMMs in an online, incremental setting where future data are not available, or might be subject to changes in statistics. So, initializing with K-means is not an option because in this case we would need to see a large chunk of data from the future. And even if we could, the initialization this would give us could be a harmful one when data statistics suddenly change.
>
> other comments: - negative log-likelihood is used in the results section. It would be good to clarify it somewhere since the paper only mentions "log-likelihood" but report a loss that should be minimized - Section 4.4 "Differently from the other experiments, we choose a learning rate of = 0.02 since the original value does not lead to convergent learning. The alternative is to double the training time which works as well." -> in the first sentence, I suppose it is more a matter of "slow learning" than "non-convergent learning"
> **RESPONSE** we fully agree and will clarify this!

---

### Official Review · AnonReviewer3 · 2019-10-25
**Official Blind Review #3**

**Rating:** 3

**Review:**

The paper proposes a new method based on Stochastic Gradient Descent to train Gaussian Mixture Models, and studies this method especially in the context of high dimension.
The method is based on optimizing max-lower bound to the log-likelihood, together with a regularization term, using stochastic gradient descent.
The method is applied to two image datasets, and seem to produce sensible results.

However, in my opinion, the results and presentation do not seem at the level suitable for publication.

There are no guarantees for convergence/other performance measures for the new method, in contrast to recent methods based on moment matching (e.g Ge, Huand and Kakade, 2015). Therefore, the new method and paper should provide excellent empirical results to be worthy of publication.

The authors write in the 'related work' section that GMM with regularization was proposed by [Verbeek et al. 2005], but it is an older idea - for example [Ormoneit&Tresp 1998]

In Section 3.1, the motivation for the maximization in eq. (2) is unclear.
Why is it easier to compute the gradient this way rather than keep the original likelihood?
Moreover, the max operation is non-smooth and can cause problems with the definition of the gradient at some points.
The authors point to a problem of underflow/overflow when evaluating the gradient of the
full log-likelihood because the densities p(x | k) can be very small - but it is standard practice in EM to keep all probabilities multiplied by say their maximum p_max and keep log(p_max) separately, to avoid underflow problems.

Section 3.2: I don't understand the requirement \Sigma_{i,j} >=0. Is it a requirement for each entry of the covariance matrix? (which covariance matrix? there are K such matrices). The requirement should be that each matrix is positive-definite, not the entries.

Section 3.3: The first sentence is wrong - EM can also suffer from the problem of local minima.
Also, the single-component solution doesn't seem like a local minimum - but rather the log-likelihood is unbounded here
since you can put another component on one of the data points with infinitely small variance.
The degenerate solution where all Gaussians have equal weights, mean and variance does not seem like a local minimum
of the log-likelihood. Say the data really comes from 2 distinct Gaussians - then separating the two Gaussians means a bit would increase the log-likelihood. is not a local minimum. I'm not even sure if the gradient is zero at this point - the authors should show this. Maybe the authors mean their modified loss L_MC - this should be stated clearly.

The change of regularization during the training process seems like a heuristic that worked well for the authors, but it is thus unclear what optimization problem is the optimization solving. The regularization is thus included for optimization reasons, and not in the usual sense of regularization.


The expression for \tau at the end of Section 3.3 seems wrong to me. I don't see how plugging it into eq. (7) gives a continuous \sigma(t) function.

Section 3.4: What is \mu^i? I didn't see a definition

I don't understand the local principal directions covariance structure. The authors write 'a diagonal covariance matrix of S < D entries). But what about the other D-S coordinates? are they all zero? or can have any values?
The parameters count lists (S+1)*D+1 for each Gaussian so I'm assuming S*D parameters are used for the covariance matrix, but it is unclear how. Eq. (9) has the parameters vectors d_{ks} for the principal directions, together with the \Sigma_ss scalar values - it would be good to relate them to the mean and variance of the Gaussians.


Section 4: The paragraph that describes the experimental details at the beginning is repeated twice.

The experimental results are not very convincing. The images in Figure 2,4 were picked by an unknown (manual?) criteria.

In the comparison to EM in Figure 3 there are missing details - which log-likelihood is used? L, L_MC? or different ones for different methods? is this test set log-likelihood? what fraction of the data was used?
There is also no comparison of running time between the two methods.




**Experience Assessment:**

I have published one or two papers in this area.

**Review Assessment: Checking Correctness Of Derivations And Theory:**

I assessed the sensibility of the derivations and theory.

**Review Assessment: Checking Correctness Of Experiments:**

I carefully checked the experiments.

**Review Assessment: Thoroughness In Paper Reading:**

I read the paper thoroughly.

---

> ### Author Response · Authors · 2019-11-15
> **Response to reviewers**
>
> Thank you for the comprehensive review, it is the first time we publish on GMMs so this feedback is invaluable. Here our responses, we could not incorporate all the changes yet but they will come!
>
> There are no guarantees for convergence/other performance measures for the new method, in contrast to recent methods based on moment matching (e.g Ge, Huand and Kakade, 2015). Therefore, the new method and paper should provide excellent empirical results to be worthy of publication.
> **RESPONSE** The guarantees for convergence come, to our mind, simply from the fact that we minimize an energy function (bounded from below) by gradient descent. If the step size is small enough, we are sure to reach at leats a local minimum and stay there, ensuring stability. It is not the same case as for EM, we one needs to explicitly show that E and M steps cannot increase the log-likelihood. The paper you mentioned is interesting, but requires ahead-of-time knowledge (and computation) of at least the fourth moments of the data, which requires batch processing. Our approach focusses on online GMM where samples arrive one by one and future samples are unknown until they arrive.
>
> The authors write in the 'related work' section that GMM with regularization was proposed by [Verbeek et al. 2005], but it is an older idea - for example [Ormoneit&Tresp 1998]
> → will be corrected.
>
> In Section 3.1, the motivation for the maximization in eq. (2) is unclear. Why is it easier to compute the gradient this way rather than keep the original likelihood? Moreover, the max operation is non-smooth and can cause problems with the definition of the gradient at some points.
> **RESPONSE** the gradient is much easier to compute and numerically more stable because the log of the maximal likelihood is a a very simple function of the model parameters, namely the argument of the exponential
> **RESPONSE** we beg to differ concerning the max-operation: it is smooth everywhere but potentially non-differentiable at a set of points of measure zero. If it is just a zero-measure set of points where the derivative is undefined, we can handle this case: the same is done for ReLU (smooth everywhere but non-differentiable at x=0)
>
>  The authors point to a problem of underflow/overflow when evaluating the gradient of the full log-likelihood because the densities p(x | k) can be very small - but it is standard practice in EM to keep all probabilities multiplied by say their maximum p_max and keep log(p_max) separately, to avoid underflow problems.
> **RESPONSE**  We are very much aware of this trick. It is ok for EM where it cures numerical instabilities by normalizing by the max. If we do this for the full gradient of the log-likelihood (which is different from the simplified EM form), the numerical instabilities are not cured at all.
>
> Section 3.2: I don't understand the requirement \Sigma_{i,j} >=0. Is it a requirement for each entry of the covariance matrix? (which covariance matrix? there are K such matrices). The requirement should be that each matrix is positive-definite, not the entries.
> **RESPONSE** true. But since all we treat here are diagonal covariance matrices anyway, their entries must be positive for positive-definiteness. We will clarify this.

---

> ### Author Response · Authors · 2019-11-15
> **Response to reviewer, part II**
>
> Section 3.3: The first sentence is wrong - EM can also suffer from the problem of local minima.
> **RESPONSE** agreed! This will be clarified.
>
> Also, the single-component solution doesn't seem like a local minimum - but rather the log-likelihood is unbounded here since you can put another component on one of the data points with infinitely small variance. The degenerate solution where all Gaussians have equal weights, mean and variance does not seem like a local minimum of the log-likelihood. Say the data really comes from 2 distinct Gaussians - then separating the two Gaussians means a bit would increase the log-likelihood. is not a local minimum. I'm not even sure if the gradient is zero at this point - the authors should show this. Maybe the authors mean their modified loss L_MC - this should be stated clearly.
> **RESPONSE** We updated the corresponsing section of the paper, giving the gradient explicitly and showing that both for degenerate and single-component solutions the gradient is zero. Which does not necessarily indicate a local minimum, but even saddle points are usually hard to get out of.
>
> The change of regularization during the training process seems like a heuristic that worked well for the authors, but it is thus unclear what optimization problem is the optimization solving. The regularization is thus included for optimization reasons, and not in the usual sense of regularization.
> **RESPONSE** you are right. Maybe “regularization” is the wrong term here: what we do is much more akin to annealing, starting with a high “temperature” (radius) and reducing it over time. We will adapt this in the whole paper.
>
> The expression for \tau at the end of Section 3.3 seems wrong to me. I don't see how plugging it into eq. (7) gives a continuous \sigma(t) function.
> **RESPONSE** We respectfully disagree: if you plug \tau into the expression \sigma_0 \exp(-t/\tau), you get an exponential that is \sigma_0 at t=t_0 and \sigma_\infty at t=t_\infty
>
> Section 3.4: What is \mu^i? I didn't see a definition I don't understand the local principal directions covariance structure. The authors write 'a diagonal covariance matrix of S < D entries). But what about the other D-S coordinates? are they all zero? or can have any values? The parameters count lists (S+1)*D+1 for each Gaussian so I'm assuming S*D parameters are used for the covariance matrix, but it is unclear how. Eq. (9) has the parameters vectors d_{ks} for the principal directions, together with the \Sigma_ss scalar values - it would be good to relate them to the mean and variance of the Gaussians.
> **RESPONSE** the \mu^i are just ad-hoc numerical parameters that define the initialization range of the prototypes, we will clarify this.
> **RESPONSE** concerning the principal directions: we still assume a diagonal covariance matrix, but instead of computing the covariances along the coordinate axes, we introduce a set of D principal directions per prototype, along which covariances are adapted. If we had D principal directions per prototype, this would mean K*D*D additional parameters. However, if we learn these principal directions using a PCA-like mechanism, we can actually ignore most of them and keep only the first S<<D of them, resulting in K*S*D parameters. The corresponding entries on the diagonal of the covariance matrix would be zero for the ignored directions. We will describe this better!
>
> Section 4: The paragraph that describes the experimental details at the beginning is repeated twice.
> **RESPONSE** oops! Thanks for pointing this out!
>
>  The experimental results are not very convincing. The images in Figure 2,4 were picked by an unknown (manual?) criteria. In the comparison to EM in Figure 3 there are missing details - which log-likelihood is used? L, L_MC? or different ones for different methods? is this test set log-likelihood? what fraction of the data was used? There is also no comparison of running time between the two methods.
> **RESPONSE** we will add the required information and add more results in the appendix. Yes, we always use test set log likelihood, and the log likelihood is the real one (ie not L_LC), otherwise it would not be a fair comparison to EM.

---

### Official Review · AnonReviewer2 · 2019-10-28
**Official Blind Review #2**

**Rating:** 1

**Review:**

The paper tackles the problem of online learning for GMMs, in the context of high-dimensional data.  Specifically, the authors limit the scope to SGD-like approaches and EM-like optimization.  They also offer a TF implementation.

I feel that this work is largely incremental, but more importantly indicating the authors' lack of understanding of the very long history of (online) EM.  While the authors do acknowledge some of the online EM work, they go on to develop what is a rather ad-hoc approach to online EM.

The max-component approximation in Sec. 3.1 is claimed to address the issue of numerical stability.  The authors do not appear to resort to the log-sum-exp "trick", which tackles such problems.  (In fact, their max approx is of this type.)

Sec. 3.2 uses a very standard representation of multinouli in terms of its natural parameters, which the authors again do not refer to.

The "smoothing" in Sec. 3.3 is hard to justify and difficult to understand, esp. the gridding approach.  Why not use hierarchical priors instead?

In Sec. 3.4, additional smoothing is accomplished using a subspace approach, which requires QR decomposition.  How will this affect computational efficiency, if the subspace needs to be recomputed?

Finally, I have strong concerns about the experimental evaluation.  The authors choose datasets where the sample ambient space is at most 28x28, which is not exactly (very) high-dimensional.

I am mostly concerned about the evaluation.

**Experience Assessment:**

I have published one or two papers in this area.

**Review Assessment: Checking Correctness Of Derivations And Theory:**

I assessed the sensibility of the derivations and theory.

**Review Assessment: Checking Correctness Of Experiments:**

I assessed the sensibility of the experiments.

**Review Assessment: Thoroughness In Paper Reading:**

I read the paper at least twice and used my best judgement in assessing the paper.

---

> ### Author Response · Authors · 2019-11-15
> **Response to reviewer**
>
> Thank you for the effort of reviewing our paper! Here out comments to your remarks:
>
>  I feel that this work is largely incremental, but more importantly indicating the authors' lack of understanding of the very long history of (online) EM. While the authors do acknowledge some of the online EM work, they go on to develop what is a rather ad-hoc approach to online EM.
> **RESPONSE** to put this very clearly: we do not address online EM here. We address Stochastic Gradient Descent, which is an, altogether different thing. This is why we mention only a few and recent works on online EM because they are somewhat related concerning their intent, but not at all in their way of achieving this.
>
> The max-component approximation in Sec. 3.1 is claimed to address the issue of numerical stability. The authors do not appear to resort to the log-sum-exp "trick", which tackles such problems. (In fact, their max approx is of this type.)
> **RESPONSE**  We are very much aware of this trick. It is ok for EM where it cures numerical instabilities by normalizing by the max. If we do this for the full gradient of the log-likelihood (which is different from the simplified EM form), the numerical instabilities are not cured at all.
>
> Sec. 3.2 uses a very standard representation of multinouli in terms of its natural parameters, which the authors again do not refer to.
> **RESPONSE** could you clarify this? Is there a specific reference you feel we should cite?
>
> The "smoothing" in Sec. 3.3 is hard to justify and difficult to understand, esp. the gridding approach. Why not use hierarchical priors instead?
> **RESPONSE** the gridding approach is borrowed from SOMs where it is very effective for ensuring convergence (even in the absence of an energy function). We should maybe better call this procedure "annealing" rather than "regularization" since that's what happens: we start training at a high "temperature" (radius), where all prototypes/centroids are forced to be similar, and later relax the temperature so differences can "crystallize out". Hierarchical priors might be useful for initially allowing large values for the variances and then over time to reduce these. Not possible to do for this article, but this will be picked up in subsequent work.
>
> In Sec. 3.4, additional smoothing is accomplished using a subspace approach, which requires QR decomposition. How will this affect computational efficiency, if the subspace needs to be recomputed?
> **RESPONSE** a fair point. We plan to replace the QR decomposition by an orthonormalization term in the gradient. This works well as long as the different principal directions are orthonormal in the beginning, which is simple to achieve.
>
> Finally, I have strong concerns about the experimental evaluation. The authors choose datasets where the sample ambient space is at most 28x28, which is not exactly (very) high-dimensional. I am mostly concerned about the evaluation.
> **RESPONSE** We chose SVHN as well, which is 32x32x3 so 3000 dimensions. Always difficult to define what high-dimensional means, but when considering what GMMs are normally applied to (2-50 dimensions at most) we believe this qualifies as high-dimensional.

---

### Decision · Program_Chairs · 2019-12-19

**Decision:**

Reject

**Comment:**

The paper presents an SGD-based learning of a Gaussian mixture model, designed to match a data streaming setting.

The reviews state that the paper contains some quite good points, such as
* the simplicity and scalability of the method, and its robustness w.r.t. the initialization of the approach;
* the SOM-like approach used to avoid degenerated solutions;

Among the weaknesses are
* an insufficient discussion wrt the state of the art, e.g. for online EM;
* the description of the approach seems yet not mature (e.g., the constraint enforcement boils down to considering that the $\pi_k$ are obtained using softmax; the discussion about the diagonal covariance matrix vs the use of local principal directions is not crystal clear);
* the fact that experiments need be strengthened.

I thus encourage the authors to rewrite and polish the paper, simplifying the description of the approach and better positioning it w.r.t. the state of the art (in particular, mentioning the data streaming motivation from the start). Also, more evidence, and a more thorough analysis thereof, must be provided to back up the approach and understand its limitations.